# VECoDeR - Variational Embeddings for Community Detection and Node Representation

## Abstract

In this paper, we study how to simultaneously learn two highly correlated tasks of graph analysis, i.e., community detection and node representation learning. We propose an efficient generative model called **VECoDeR** for jointly learning **V**ariational **E**mbeddings for **Co**mmunity **De**tection and node **R**epresentation. VECoDeR assumes that every node can be a member of one or more communities. The node embeddings are learned in such a way that connected nodes are not only "closer" to each other but also share similar community assignments. A joint learning framework leverages community-aware node embeddings for better community detection. We demonstrate on several graph datasets that VECoDeR effectively outperforms many competitive baselines on all three tasks i.e. node classification, overlapping community detection and non-overlapping community detection. We also show that VECoDeR is computationally efficient and has quite robust performance with varying hyperparameters.

## 1 Introduction

Graphs are flexible data structures that model complex relationships among entities, i.e. data points as nodes and the relations between nodes via edges. One important task in graph analysis is community detection, where the objective is to cluster nodes into multiple groups (communities). Each community is a set of densely connected nodes. The communities can be overlapping or non-overlapping, depending on whether they share some nodes or not. Several algorithmic (Ahn et al., 2010; Derényi et al., 2005) and probabilistic approaches (Gopalan & Blei, 2013; Leskovec & Mcauley, 2012; Wang et al., 2017; Yang et al., 2013) to community detection have been proposed. Another fundamental task in graph analysis is learning the node embeddings. These embeddings can then be used for downstream tasks like graph visualization (Tang et al., 2016; Wang et al., 2016; Gao et al., 2011; Wang et al., 2017) and classification (Cao et al., 2015; Tang et al., 2015).

In the literature, these tasks are usually treated separately. Although the standard graph embedding methods capture the basic connectivity, the learning of the node embeddings is independent of community detection. For instance, a simple approach can be to get the node embeddings via DeepWalk (Perozzi et al., 2014) and get community assignments for each node by using k-means or Gaussian mixture model. Looking from the other perspective, methods like Bigclam (Yang & Leskovec (2013)), that focus on finding the community structure in the dataset, perform poorly for node-representation tasks e.g. node classification. This motivates us to study the approaches that jointly learn community-aware node embeddings.

Recently several approaches, like CNRL (Tu et al., 2018), ComE (Cavallari et al., 2017), vGraph (Sun et al. (2019)) etc, have been proposed to learn the node embeddings and detect communities simultaneously in a unified framework. Several studies have shown that community detection is improved by incorporating the node representation in the learning process (Cao et al., 2015; Kozdoba & Mannor, 2015). The intuition is that the global structure of graphs learned during community detection can provide useful context for node embeddings and vice versa.

The joint learning methods (CNRL, ComE and vGraph) learn two embeddings for each node. One node embedding is used for the node representation task. The second node embedding is the "context" embedding of the node which aids in community detection. As CNRL and ComE are based on Skip-Gram (Mikolov et al., 2013) and DeepWalk (Perozzi et al., 2014), they inherit "context" embedding from it for learning the neighbourhood information of the node. vGraph also requires two

node embeddings for parameterizing two different distributions. In contrast, we propose learning a single community-aware node representation which is directly used for both tasks. In this way, we not only get rid of an extraneous node embedding but also reduce the computational cost.

In this paper, we propose an efficient generative model called **VECoDeR** for jointly learning both community detection and node representation. The underlying intuition behind VECoDeR is that every node can be a member of one or more communities. However, the node embeddings should be learned in such a way that connected nodes are "closer" to each other than unconnected nodes. Moreover, connected nodes should have similar community assignments. Formally, we assume that for $i$-th node, the node embeddings $z_i$ are generated from a prior distribution $p(z)$. Given $z_i$, the community assignments $c_i$ are sampled from $p(c_i|z_i)$, which is parameterized by node and community embeddings. In order to generate an edge $(i, j)$, we sample another node embedding $z_j$ from $p(z)$ and respective community assignment $c_j$ from $p(c_j|z_j)$. Afterwards, the node embeddings and the respective community assignments of node pairs are fed to a decoder. The decoder ensures that embeddings of both the nodes and the communities of connected nodes share high similarity. This enables learning such node embeddings that are useful for both community detection and node representation tasks.

We validate the effectiveness of our approach on several real-world graph datasets. In Sec. 4, we show empirically that VECoDeR is able to outperform the baseline methods including the direct competitors on all three tasks i.e. node classification, overlapping community detection and non-overlapping community detection. Furthermore, we compare the computational cost of training different algorithms. VECoDeR is up to 40x more time-efficient than its competitors. We also conduct hyperparameter sensitivity analysis which demonstrates the robustness of our approach. Our main contributions are summarized below:

- We propose an efficient generative model called **VECoDeR** for joint community detection and node representation learning.
- We adopt a novel approach and argue that a single node embedding is sufficient for learning both the representation of the node itself and its context.
- Training VECoDeR is extremely time-efficient in comparison to its competitors.

## 2 RELATED WORK

**Community Detection.** Early community detection algorithms are inspired from clustering algorithms (Xie et al., 2013). For instance, spectral clustering (Tang & Liu, 2011) is applied to the graph Laplacian matrix for extracting the communities. Similarly, several matrix factorization based methods have been proposed to tackle the community detection problem. For example, Bigclam (Yang & Leskovec (2013)) treats the problem as a non-negative matrix factorization (NMF) task. It aims to recover the node-community affiliation matrix and learns the latent factors which represent community affiliations of nodes. Another method CESNA(Yang et al. (2013)) extends Bigclam by modelling the interaction between the network structure and the node attributes. The performance of matrix factorization methods is limited due to the capacity of the bi-linear models. Some generative models, like vGraph (Sun et al., 2019), Circles (Leskovec & Mcauley (2012)) etc, have also been proposed to detect communities in a graph.

**Node Representation Learning.** Many successful algorithms which learn node representation in an unsupervised way are based on random walk objectives (Perozzi et al., 2014; Tang et al., 2015; Grover & Leskovec, 2016; Hamilton et al., 2017). Some known issues with random-walk based methods (e.g. DeepWalk, node2vec etc) are: (1) They sacrifice the structural information of the graph by putting over-emphasis on the proximity information (Ribeiro et al., 2017) and (2) great dependence of the performance on hyperparameters (walk-length, number of hops etc) (Perozzi et al., 2014; Grover & Leskovec, 2016). Recently, Gilmer et al. (2017) recently showed that graph convolutions encoder models greatly reduce the need for using the random-walk based training objectives. This is because the graph convolutions enforce that the neighboring nodes have similar representations. Some interesting GCN based approaches include graph autoencoders e.g. GAE and VGAE(Kipf & Welling (2016b)) and DGI(Velickovic et al., 2019).

**Joint community detection and node representation learning.** In the literature, several attempts have been made to tackle both these tasks in a single framework. Most of these methods propose

an alternate optimization process, i.e. learn node embeddings and improve community assignments with them and vice versa (Cavallari et al., 2017; Tu et al., 2018). Some approaches, like CNRL (Tu et al., 2018) and ComE (Cavallari et al., 2017), are inspired from random walk, thus inheriting the shortcomings of random walk. Others, like GEMSEC (Rozemberczki et al. (2019), are limited to the detection of non-overlapping communities. There also exist some generative models like CommunityGAN (Jia et al. (2019)) and vGraph (Sun et al. (2019)) that jointly learn community assignments and node embeddings. Some methods have high computational complexity, i.e. quadratic to the number of nodes in a graph, e.g. M-NMF (Wang et al. (2017)) and DNR (Yang et al., 2016a). CNRL, ComE and vGraph require learning two embeddings for each node for simultaneously tackling the two tasks. Unlike them, VECoDeR learns a single community-aware node representation which is directly used for both tasks.

It is pertinent to highlight that although both vGraph and VECoDeR adopt a variational approach but the underlying models are quite different. vGraph assumes that each node can be represented as a mixture of multiple communities and is described by a multinomial distribution over communities, whereas VECoDeR models the node embedding by a single distribution. For a given node, vGraph, first draws a community assignment and then a connected neighbor node is generated based on the assignment. Whereas, VECoDeR draws the node embedding from prior distribution and then community assignment is conditioned on a single node only. In simple terms, vGraph also needs edge information in the generative process whereas VECoDeR does not require it. VECoDeR relies on the decoder to ensure that embeddings of the connected nodes and their communities share high similarity with each other.

## 3 METHODOLOGY

### 3.1 PROBLEM FORMULATION

Suppose an undirected graph $\mathcal{G} = (\mathcal{V}, \mathcal{E})$ with the adjacency matrix $\boldsymbol{A} \in \mathbb{R}^{N \times N}$ and a matrix $\boldsymbol{X} \in \mathbb{R}^{N \times F}$ of $F$-dimensional node features, $N$ being the number of nodes. Given $K$ as the number of communities, we aim to jointly learn the node embeddings and the community embeddings following a variational approach such that: (1) One or more communities can be assigned to every node and (2) the node embeddings can be used for both community detection and node classification.

### 3.2 VARIATIONAL MODEL

**Generative Model:** Let us denote the latent node embedding and community assignment for $i$-th node by the random variables $\boldsymbol{z}_i \in \mathbb{R}^d$ and $c_i$ respectively. The generative model is given by:

$$p(\boldsymbol{A}) = \int \sum_{\boldsymbol{c}} p(\boldsymbol{Z}, \boldsymbol{c}, \boldsymbol{A}) d\boldsymbol{Z}, \tag{1}$$

where $\boldsymbol{c} = [c_1, c_2, \cdots, c_N]$ and the matrix $\boldsymbol{Z} = [\boldsymbol{z}_1, \boldsymbol{z}_2, \cdots, \boldsymbol{z}_N]$ stacks the node embeddings. The joint distribution in (1) is mathematically expressed as

$$p(\boldsymbol{Z}, \boldsymbol{c}, \boldsymbol{A}) = p(\boldsymbol{Z}) \, p_\theta(\boldsymbol{c}|\boldsymbol{Z}) \, p_\theta(\boldsymbol{A}|\boldsymbol{c}, \boldsymbol{Z}), \tag{2}$$

where $\theta$ denotes the model parameters. Let us denote elements of $\boldsymbol{A}$ by $a_{ij}$. Following existing approaches (Kipf & Welling, 2016b; Khan et al., 2020), we consider $\boldsymbol{z}_i$ to be $i.i.d$ random variables. Furthermore, assuming $c_i|\boldsymbol{z}_i$ to be $i.i.d$ random variables, the joint distributions in (2) can be factorized as

$$p(\boldsymbol{Z}) = \prod_{i=1}^{N} p(\boldsymbol{z}_i) \quad (3) \qquad p_\theta(\boldsymbol{c}|\boldsymbol{Z}) = \prod_{i=1}^{N} p_\theta(c_i|\boldsymbol{z}_i) \quad (4)$$

$$p_\theta(\boldsymbol{A}|\boldsymbol{c}, \boldsymbol{Z}) = \prod_{i,j} p_\theta(a_{ij}|c_i, c_j, \boldsymbol{z}_i, \boldsymbol{z}_j), \tag{5}$$

where Eq. (5) assumes that the *edge decoder* $p_\theta(a_{ij}|c_i, c_j, \boldsymbol{z}_i, \boldsymbol{z}_j)$ depends only on $c_i, c_j, \boldsymbol{z}_i$ and $\boldsymbol{z}_j$.

**Inference Model:** We aim to learn the model parameters $\theta$ such that $\log(p_\theta(\boldsymbol{A}))$ is maximized. In order to ensure computational tractability, we introduce the approximate posterior

$$q_\phi(\boldsymbol{Z}, \boldsymbol{c}|\mathcal{I}) = \prod_i q_\phi(\boldsymbol{z}_i, c_i|\mathcal{I}) = \prod_i q_\phi(\boldsymbol{z}_i|\mathcal{I})q_\phi(c_i|\boldsymbol{z}_i, \mathcal{I}), \tag{6}$$

where $\mathcal{I} = (\boldsymbol{A}, \boldsymbol{X})$ if node features are available, otherwise $\mathcal{I} = \boldsymbol{A}$. We maximize the corresponding ELBO bound (for derivation, refer to the supplementary material), given by

$$\mathcal{L}_{ELBO} \approx -\sum_{i=1}^{N} D_{KL}(q_\phi(\boldsymbol{z}_i|\mathcal{I}) \,||\, p(\boldsymbol{z}_i)) - \sum_{i=1}^{N} \frac{1}{M} \sum_{m=1}^{M} D_{KL}(q_\phi(c_i|\boldsymbol{z}_i^{(m)}, \mathcal{I}) \,||\, p_\theta(c_i|\boldsymbol{z}_i^{(m)}))$$

$$+ \sum_{(i,j)\in\mathcal{E}} \mathbb{E}_{(\boldsymbol{z}_i, \boldsymbol{z}_j, c_i, c_j)\sim q_\phi(\boldsymbol{z}_i, \boldsymbol{z}_j, c_i, c_j|\mathcal{I})} \left\{ \log\left( p_\theta(a_{ij}|c_i, c_j, \boldsymbol{z}_i, \boldsymbol{z}_j) \right) \right\}, \tag{7}$$

where $D_{KL}(.||.)$ represents the KL-divergence between two distributions. The distribution $q_\phi(\boldsymbol{z}_i, \boldsymbol{z}_j, c_i, c_j|\mathcal{I})$ in the third term of Eq. (7) is factorized into two conditionally independent distributions i.e.

$$q_\phi(\boldsymbol{z}_i, \boldsymbol{z}_j, c_i, c_j|\mathcal{I}) = q_\phi(\boldsymbol{z}_i, c_i|\mathcal{I})q_\phi(\boldsymbol{z}_j, c_j|\mathcal{I}). \tag{8}$$

### 3.3 DESIGN CHOICES

In Eq. (3), $p(\boldsymbol{z}_i)$ is chosen to be the standard gaussian distribution for all $i$. The corresponding approximate posterior $q_\phi(\boldsymbol{z}_i|\mathcal{I})$ in Eq. (6), used as node embeddings encoder, is given by

$$q_\phi(\boldsymbol{z}_i|\mathcal{I}) = \mathcal{N}\big(\boldsymbol{\mu}_i(\mathcal{I}), \mathrm{diag}(\boldsymbol{\sigma}^2{}_i(\mathcal{I}))\big). \tag{9}$$

The parameters of $q_\phi(\boldsymbol{z}_i|\mathcal{I})$ can be learnt by any encoder network e.g. graph convolutional network (Kipf & Welling (2016a)), graph attention network ( Veličković et al. (2017)), GraphSAGE (Hamilton et al. (2017)) or even two matrices to learn $\boldsymbol{\mu}_i(\mathcal{I})$ and $\mathrm{diag}(\boldsymbol{\sigma}^2{}_i(\mathcal{I}))$. Samples are then generated using reparametrization trick (Doersch (2016)).

For parameterizing $p_\theta(c_i|\boldsymbol{z}_i)$ in Eq. (4), we introduce community embeddings $\{\boldsymbol{g}_1, \cdots, \boldsymbol{g}_K\}$; $\boldsymbol{g}_k \in \mathbb{R}^d$. The distribution $p_\theta(c_i|\boldsymbol{z}_i)$ is then modelled as the softmax of dot products of $\boldsymbol{z}_i$ with $\boldsymbol{g}_k$, i.e.

$$p_\theta(c_i = k|\boldsymbol{z}_i) = \frac{\exp(<\boldsymbol{z}_i, \boldsymbol{g}_k>)}{\sum\limits_{\ell=1}^{K} \exp(<\boldsymbol{z}_i, \boldsymbol{g}_\ell>)}. \tag{10}$$

The corresponding approximate posterior $q_\phi(c_i = k|\boldsymbol{z}_i, \mathcal{I})$ in Eq. (6) is affected by the node embedding $\boldsymbol{z}_i$ as well as the neighborhood. To design this, our intuition is to consider the similarity of $\boldsymbol{g}_k$ with the embedding $\boldsymbol{z}_i$ as well as with the embeddings of the neighbors of the $i$-th node. The overall similarity with neighbors is mathematically formulated as the average of the dot products of their embeddings. Afterwards a hyperparameter $\alpha$ is introduced to control the bias between the effect of $\boldsymbol{z}_i$ and the set $\mathcal{N}_i$ of the neighbors of the $i$-th node. Finally, a softmax is applied as follows

$$q_\phi(c_i = k|\boldsymbol{z}_i, \mathcal{G}) = \frac{\exp\Big( \alpha <\boldsymbol{z}_i, \boldsymbol{g}_k> + (1-\alpha)\frac{1}{|\mathcal{N}_i|} \sum\limits_{j\in\mathcal{N}_i} <\boldsymbol{z}_j, \boldsymbol{g}_k> \Big)}{\sum\limits_{\ell=1}^{K} \exp\Big( \alpha <\boldsymbol{z}_i, \boldsymbol{g}_\ell> + (1-\alpha)\frac{1}{|\mathcal{N}_i|} \sum\limits_{j\in\mathcal{N}_i} <\boldsymbol{z}_j, \boldsymbol{g}_\ell> \Big)}. \tag{11}$$

Hence, Eq. (11) ensures that graph structure information is employed to learn community assignments instead of relying on an extraneous node embedding as done in (Sun et al., 2019; Cavallari et al., 2017). Finally, the choice of edge decoder in Eq. (5) is motivated by the intuition that the nodes connected by edges have a high probability of belonging to the same community and vice versa. Therefore we model the edge decoder as:

$$p_\theta(a_{ij}|c_i = \ell, c_j = m, \boldsymbol{z}_i, \boldsymbol{z}_j) = \frac{\sigma(<\boldsymbol{z}_i, \boldsymbol{g}_m>) + \sigma(<\boldsymbol{z}_j, \boldsymbol{g}_\ell>)}{2}. \tag{12}$$

For better reconstructing the edges, Eq. (12) makes use of the community embeddings, node embeddings and community assignment information simultaneously. This helps in learning better node representations by leveraging the global information about the graph structure via community detection. On the other hand, this also forces the community assignment information to exploit the local graph structure via node embeddings and edge information.

### 3.4 PRACTICAL ASPECTS

The third term in Eq. (7) is estimated in practice using the samples generated by the approximate posterior. This term is equivalent to the negative of binary cross-entropy (BCE) loss between observed edges and reconstructed edges. Since community assignment follows a categorical distribution, we use Gumbel-softmax (Jang et al. (2016)) for backpropagation of the gradients. As for the second term of Eq. (7), it is also enough to set $M = 1$, i.e. use only one sample per input node.

For inference, non-overlapping community assignment can be obtained for $i$-th node as

$$\mathcal{C}_i = \underset{k \in \{1, \cdots, K\}}{\arg \max} \ q_\phi(c_i = k | \mathbf{z}_i, \mathcal{I}). \tag{13}$$

To get overlapping community assignments for $i$-th node, we can threshold its weighted probability vector at $\epsilon$, a hyperparameter, as follows

$$\mathcal{C}_i = \left\{ k \ \middle| \ \frac{q_\phi(c_i = k | \mathbf{z}_i, \mathcal{I})}{\max_\ell q_\phi(c_i = \ell | \mathbf{z}_i, \mathcal{I})} \geq \epsilon \right\}, \quad \epsilon \in [0, 1]. \tag{14}$$

### 3.5 COMPLEXITY

Computation of dot products for all combinations of node and community embeddings takes $\mathcal{O}(NKd)$ time. Solving Eq. (11) further requires calculation of mean of dot products over the neighborhood for every node, which takes $\mathcal{O}(|\mathcal{E}|K)$ computations overall as we traverse every edge for every community. Finally, we need softmax over all communities for every node in Eq. (10) and Eq. (11) which takes $\mathcal{O}(NK)$ time. Eq. (12) takes $\mathcal{O}(|\mathcal{E}|)$ time for all edges as we have already calculated the dot products. As a result, the overall complexity becomes $\mathcal{O}(|\mathcal{E}|K + NKd)$. This complexity is quite low compared to other algorithms designed to achieve similar goals (Cavallari et al., 2017; Wang et al., 2017; Yang et al., 2016a).

## 4 EXPERIMENTS

### 4.1 DATASETS

We have selected 18 different datasets ranging from 270 to 126,842 edges. For non-overlapping community detection and node classification, we use 5 the citation datasets (Bojchevski & Günnemann (2017); Yang et al. (2016b)). The remaining datasets (Leskovec & Mcauley (2012); Yang & Leskovec (2015)), used for overlapping community detection, are taken from SNAP repository (Leskovec & Krevl (2014)). Following (Sun et al., 2019), we take 5 biggest ground truth communities for youtube, amazon and dblp. Moreover, we also analyse the case of large number of communities. For this purpose, we prepare two subsets of amazon dataset by randomly selecting 500 and 1000 communities from 2000 smallest communities in the amazon dataset.

| Dataset | $|\mathcal{V}|$ | $|\mathcal{E}|$ | $K$ | $|F|$ | Overlap |
|---|---|---|---|---|---|
| CiteSeer | 3327 | 9104 | 6 | 3703 | N |
| CiteSeer-full | 4230 | 10674 | 6 | 602 | N |
| Cora | 2708 | 10556 | 7 | 1433 | N |
| Cora-ML | 2995 | 16316 | 7 | 2879 | N |
| Cora-full | 19793 | 126842 | 70 | 8710 | N |
| fb0 | 333 | 2519 | 24 | N/A | Y |
| fb107 | 1034 | 26749 | 9 | N/A | Y |
| fb1684 | 786 | 14024 | 17 | N/A | Y |
| fb1912 | 747 | 30025 | 46 | N/A | Y |
| fb3437 | 534 | 4813 | 32 | N/A | Y |
| fb348 | 224 | 3192 | 14 | N/A | Y |
| fb414 | 150 | 1693 | 7 | N/A | Y |
| fb698 | 61 | 270 | 13 | N/A | Y |
| youtube | 5346 | 24121 | 5 | N/A | Y |
| amazon | 794 | 2109 | 5 | N/A | Y |
| amazon500 | 1113 | 3496 | 500 | N/A | Y |
| amazon1000 | 1540 | 4488 | 1000 | N/A | Y |
| dblp | 24493 | 89063 | 5 | N/A | Y |

Table 1: Every dataset has $|\mathcal{V}|$ nodes, $|\mathcal{E}|$ edges, $K$ communities and $|F|$ features. $|F|$ = N/A means that either the features were missing or not used.

## 4.2 BASELINES

For overlapping community detection, we compare with the following competitive baselines: **MNMF**(Wang et al., 2017) learns community membership distribution by using joint non-negative matrix factorization with modularity based regularization. **BIGCLAM**(Yang & Leskovec (2013)) also formulates community detection as a non-negative matrix factorization (NMF) task. It simultaneously optimizes the model likelihood of observed links and learns the latent factors which represent community affiliations of nodes. **CESNA** (Yang et al. (2013)) extends BIGCLAM by statistically modelling the interaction between the network structure and the node attributes. **Circles** (Leskovec & Mcauley (2012)) introduces a generative model for community detection in ego-networks by learning node similarity metrics for every community. **SVI** (Gopalan & Blei (2013)) formulates membership of nodes in multiple communities by a Bayesian model of networks. **vGraph** (Sun et al. (2019)) simultaneously learns node embeddings and community assignments by modelling the nodes as being generated from a mixture of communities. **vGraph+**, a variant further incorporates regularization to weigh local connectivity. **ComE** (Cavallari et al. (2017)) jointly learns community and node embeddings by using gaussian mixture model formulation. **CNRL**(Tu et al., 2018) enhances the random walk sequences (generated by DeepWalk, node2vec etc) to jointly learn community and node embeddings. **CommunityGAN** (ComGAN)is a generative adversarial model for learning node embeddings such that the entries of the embedding vector of each node refer to the membership strength of the node to different communities. Lastly, we compare the results with the communities obtained by applying k-means to the learned embeddings of **DGI** (Velickovic et al., 2019).

For non-overlapping community detection and node classification, in addition to MNMF, DGI, CNRL, CommunityGAN, vGraph and ComE, we compare VECODER with the following baselines: **DeepWalk** (Perozzi et al. (2014)) makes use of SkipGram (Mikolov et al. (2013)) and truncated random walks on network to learn node embeddings. **LINE** (Tang et al. (2015)) learns node embeddings while attempting to preserve first and second order proximities of nodes. **Node2Vec** (Grover & Leskovec (2016)) learns the embeddings using biased random walk while aiming to preserve network neighborhoods of nodes. **Graph Autoencoder (GAE)**Kipf & Welling (2016b) extends the idea of autoencoders to graph datasets. We also include its variational counterpart i.e. **VGAE**. **GEMSEC** is a sequence sampling-based learning model which aims to jointly learn the node embeddings and clustering assignments.

## 4.3 SETTINGS

**For overlapping community detection**, we learn mean and log-variance matrices of 16-dimensional node embeddings. We set $\alpha = 0.9$ and $\epsilon = 0.3$ in all our experiments. Following Kipf & Welling (2016b), we first pre-train a variational graph autoencoder. We perform gradient descent with Adam optimizer (Kingma & Ba (2014)) and learning rate $= 0.01$. Community assignments are obtained using Eq. (14). For the baselines, we employ the results reported by Sun et al. (2019). For evaluating the performance, we use *F1-score* and *Jaccard similarity*.

**For non-overlapping community detection**, since the default implementations of most the baselines use 128 dimensional embeddings, for we use $d = 128$ for fair comparison. Eq. (13) is used for community assignments. For vGraph, we use the code provided by the authors. We employ *normalized mutual information (NMI)* and *adjusted random index (ARI)* as evaluation metrics.

**For node classification**, we follow the training split used in various previous works (Yang et al., 2016b; Kipf & Welling, 2016a; Velickovic et al., 2019), i.e. 20 nodes per class for training. We train logistic regression using LIBLINEAR (Fan et al. (2008)) solver as our classifier and report the evaluation results on rest of the nodes. For the algorithms that do not use node features, we train the classifier by appending the raw node features with the learnt embeddings. For evaluation, we use *F1-macro* and *F1-micro* scores.

All the reported results are the average over five runs. Further implementation details can be found in the code: https://anonymous.4open.science/r/1d95bf8f-8ce3-4870-a454-07db463b419f.

## 4.4 DISCUSSION OF RESULTS

In the following, we discuss the results to gain some important insights into the problem.

| F1 Score (%) | | | | | | | | | | | |
|---|---|---|---|---|---|---|---|---|---|---|---|
| **Dataset** | **MNMF** | **Bigclam** | **CESNA** | **Circles** | **SVI** | **vGraph** | **vGraph+** | **ComE** | **CNRL** | **ComGan** | **DGI** | **VECODER** |
| **fb0** | 14.4 | 29.5 | 28.1 | 28.6 | 28.1 | 24.4 | 26.1 | 31.1 | 11.5 | **35.0** | 27.35 | 34.7 |
| **fb107** | 12.6 | 39.3 | 37.3 | 24.7 | 26.9 | 28.2 | 31.8 | 39.7 | 20.2 | 47.5 | 35.78 | **59.7** |
| **fb1684** | 12.2 | 50.4 | 51.2 | 28.9 | 35.9 | 42.3 | 43.8 | 52.9 | 38.5 | 47.6 | 42.85 | **56.4** |
| **fb1912** | 14.9 | 34.9 | 34.7 | 26.2 | 28.0 | 25.8 | 37.5 | 28.7 | 8.0 | 35.6 | 32.6 | **45.8** |
| **fb3437** | 13.7 | 19.9 | 20.1 | 10.1 | 15.4 | 20.9 | 22.7 | 21.3 | 3.9 | 39.3 | 19.66 | **50.2** |
| **fb348** | 20.0 | 49.6 | 53.8 | 51.8 | 46.1 | 55.4 | 53.1 | 46.2 | 34.1 | 55.8 | 54.68 | **58.2** |
| **fb414** | 22.1 | 58.9 | 60.1 | 48.4 | 38.9 | 64.7 | 66.9 | 55.3 | 25.3 | 43.9 | 56.93 | **69.6** |
| **fb698** | 26.6 | 54.2 | 58.7 | 35.2 | 40.3 | 54.0 | 59.5 | 45.8 | 16.4 | 58.2 | 52.19 | **64.0** |
| **Youtube** | 59.9 | 43.7 | 38.4 | 36.0 | 41.4 | 50.7 | 52.2 | 65.5 | 51.4 | 43.6 | 47.8 | **67.3** |
| **Amazon** | 38.2 | 46.4 | 46.8 | 53.3 | 47.3 | 53.3 | 53.2 | 50.1 | 53.5 | 51.4 | 44.7 | **58.1** |
| **Amazon500** | 30.1 | 52.2 | 57.3 | 46.2 | 41.9 | 61.2 | 60.4 | 59.8 | 38.4 | 59.3 | 33.8 | **67.6** |
| **Amazon1000** | 19.3 | 28.6 | 30.8 | 25.9 | 21.6 | 54.3 | 47.3 | 50.3 | 27.1 | 52.7 | 37.7 | **60.5** |
| **Dblp** | 21.8 | 23.6 | 35.9 | 36.2 | 33.7 | 39.3 | 39.9 | 47.1 | 46.8 | 34.9 | 44 | **53.9** |

Table 2: F1 scores for overlapping communities. Best and second best values are bold and blue respectively.

| jaccard (%) | | | | | | | | | | | |
|---|---|---|---|---|---|---|---|---|---|---|---|
| **Dataset** | **MNMF** | **Bigclam** | **CESNA** | **Circles** | **SVI** | **vGraph** | **vGraph+** | **ComE** | **CNRL** | **ComGan** | **DGI** | **VECODER** |
| **fb0** | 8.0 | 18.5 | 17.3 | 18.6 | 17.6 | 14.6 | 15.9 | 19.5 | 6.8 | 24.1 | 16.8 | **24.7** |
| **fb107** | 6.9 | 27.5 | 27.0 | 15.5 | 17.2 | 18.3 | 21.7 | 28.7 | 11.9 | 38.5 | 25.29 | **46.8** |
| **fb1684** | 6.6 | 38.0 | 38.7 | 18.7 | 24.7 | 29.2 | 32.7 | 40.3 | 25.8 | 37.9 | 38.85 | **42.5** |
| **fb1912** | 8.4 | 24.1 | 23.9 | 16.7 | 20.1 | 18.6 | 28.0 | 18.5 | 4.6 | 13.5 | 22.48 | **37.3** |
| **fb3437** | 7.7 | 11.5 | 11.7 | 5.5 | 9.0 | 12.0 | 13.3 | 12.5 | 2.0 | 33.4 | 11.55 | **36.2** |
| **fb348** | 11.3 | 35.9 | 40.0 | 39.3 | 33.6 | 41.0 | 40.5 | 34.4 | 21.7 | 23.2 | 41.77 | **43.5** |
| **fb414** | 12.8 | 47.1 | 47.3 | 34.2 | 29.3 | 51.8 | 55.9 | 42.2 | 15.4 | 53.6 | 46.36 | **58.4** |
| **fb698** | 16.0 | 41.9 | 45.9 | 22.6 | 30.0 | 43.6 | 47.7 | 33.8 | 9.6 | 46.9 | 42.1 | **50.4** |
| **Youtube** | 46.7 | 29.3 | 24.2 | 22.1 | 28.7 | 34.3 | 34.8 | 52.5 | 35.5 | 44.0 | 32.72 | **53.3** |
| **Amazon** | 25.2 | 35.1 | 35.0 | 36.7 | 36.4 | 36.9 | 36.9 | 34.6 | 38.7 | 38.0 | 29.09 | **41.9** |
| **Amazon500** | 20.8 | 51.2 | 53.8 | 47.2 | 45.0 | 59.1 | 59.6 | 58.4 | 41.1 | 57.3 | 23.28 | **64.9** |
| **Amazon1000** | 20.3 | 26.8 | 28.9 | 24.9 | 23.6 | 54.3 | 49.7 | 52.0 | 26.9 | 54.1 | 23.25 | **57.1** |
| **Dblp** | 20.9 | 13.8 | 22.3 | 23.3 | 20.9 | 25.0 | 25.1 | 27.9 | 32.8 | 25.0 | 29.15 | **37.3** |

Table 3: Jaccard scores for overlapping communities. Best and second best values are bold and blue respectively.

| | NMI(%) | | | | | ARI(%) | | | | |
|---|---|---|---|---|---|---|---|---|---|---|
| **Algorithm** | **CiteSeer** | **CiteSeer-full** | **Cora** | **Cora-ML** | **Cora-full** | **CiteSeer** | **CiteSeer-full** | **Cora** | **Cora-ML** | **Cora-full** |
| **MNMF** | 14.1 | 9.4 | 19.7 | 37.8 | 42.0 | 2.6 | 0.4 | 2.9 | 24.1 | 6.1 |
| **DeepWalk** | 8.8 | 15.4 | 39.7 | 43.2 | 48.5 | 9.5 | 16.4 | 31.2 | 33.9 | 22.5 |
| **LINE** | 8.7 | 13.0 | 32.8 | 42.3 | 40.3 | 3.3 | 3.7 | 14.9 | 32.7 | 11.7 |
| **Node2Vec** | 14.9 | 22.3 | 39.7 | 39.6 | 48.1 | 8.1 | 10.5 | 25.8 | 27.9 | 18.8 |
| **GAE** | 17.4 | 55.1 | 39.7 | 48.3 | 48.3 | 14.1 | 50.6 | 29.3 | 41.8 | 18.3 |
| **VGAE** | 16.3 | 48.4 | 40.8 | 48.3 | 47.0 | 10.1 | 40.6 | 34.7 | 42.5 | 17.9 |
| **DGI** | 37.8 | 56.7 | 50.1 | 46.2 | 39.9 | 38.1 | 50.8 | 44.7 | 42.1 | 12.1 |
| **GEMSEC** | 11.8 | 11.1 | 27.4 | 18.1 | 10.0 | 0.6 | 1.0 | 4.8 | 1.0 | 0.2 |
| **CNRL** | 13.6 | 23.3 | 39.4 | 42.9 | 47.7 | 12.8 | 20.2 | 31.9 | 32.5 | 22.9 |
| **ComGAN** | 3.2 | 16.2 | 5.7 | 11.5 | 15.0 | 1.2 | 4.9 | 3.2 | 6.7 | 0.6 |
| **vGraph** | 9.0 | 7.6 | 26.4 | 29.8 | 41.7 | 5.1 | 4.2 | 12.7 | 21.6 | 14.9 |
| **ComE** | 18.8 | 32.8 | 39.6 | 47.6 | 51.2 | 13.8 | 20.9 | 34.2 | 37.2 | 19.7 |
| **VECODER** | **38.5** | **59.0** | **52.7** | **56.3** | **55.2** | 35.2 | **60.3** | **45.1** | **49.8** | **28.8** |

Table 4: Non-overlapping community detection results. Best and second best values are bold and blue respectively.

| | F1-macro(%) | | | | | F1-micro(%) | | | | |
|---|---|---|---|---|---|---|---|---|---|---|
| **Algorithm** | **CiteSeer** | **CiteSeer-full** | **Cora** | **Cora-ML** | **Cora-full** | **CiteSeer** | **CiteSeer-full** | **Cora** | **Cora-ML** | **Cora-full** |
| **MNMF** | 57.4 | 68.6 | 60.9 | 64.2 | 30.4 | 60.8 | 68.1 | 62.7 | 64.2 | 32.9 |
| **DeepWalk** | 49.0 | 56.6 | 69.7 | 75.8 | 41.7 | 52.0 | 57.3 | 70.2 | 75.6 | 48.3 |
| **LINE** | 55.0 | 60.2 | 68.0 | 75.3 | 39.4 | 57.7 | 60.0 | 68.3 | 74.6 | 42.1 |
| **Node2Vec** | 55.2 | 61.0 | 71.3 | 78.4 | 42.3 | 57.8 | 61.5 | 71.4 | 78.6 | 48.1 |
| **GAE** | 57.9 | 79.9 | 71.2 | 76.5 | 36.6 | 61.6 | 79.6 | 73.5 | 77.6 | 41.8 |
| **VGAE** | 59.1 | 74.4 | 70.4 | 75.2 | 32.4 | 62.2 | 74.4 | 72.0 | 76.4 | 37.7 |
| **DGI** | 62.6 | 82.1 | 71.1 | 72.6 | 16.5 | 67.9 | 81.8 | 73.3 | 75.4 | 21.1 |
| **GEMSEC** | 37.5 | 53.3 | 60.3 | 70.6 | 35.8 | 39.4 | 53.5 | 59.4 | 72.5 | 38.9 |
| **CNRL** | 50.0 | 58.0 | 70.4 | 77.8 | 41.3 | 53.2 | 57.9 | 70.4 | 78.4 | 45.9 |
| **ComGAN** | 55.9 | 65.7 | 56.6 | 62.5 | 27.7 | 59.1 | 64.9 | 58.5 | 62.8 | 29.4 |
| **vGraph** | 30.8 | 28.5 | 44.7 | 59.8 | 33.4 | 32.1 | 28.5 | 44.6 | 62.3 | 37.6 |
| **ComE** | 59.6 | 69.9 | 71.6 | 78.5 | 42.2 | 63.1 | 70.2 | 74.2 | 79.5 | 47.8 |
| **VECODER** | **64.8** | 76.8 | **73.1** | **80.2** | **43.1** | 68.2 | 77.0 | **75.6** | 82.0 | 49.6 |

Table 5: Node classification results. Best and second best values are bold and blue respectively.

Tables 2 and 3 summarize the results of the performance comparison for the overlapping community detection task.

First, we note that our proposed method VECODER outperforms the competitive methods on all datasets in terms of Jaccard Similarity. VECODER also outperforms its competitors on 12 out of 13 datasets in terms of F1-score. It is the second best method on the 13th dataset (*fb0*). These results demonstrate the capability of VECODER to learn multiple community assignments quite well and hence reinforces our intuition behind the design of Eq. (11).

Second, we observe that there is no consistent performing algorithm among the competitive methods. That is, excluding VECODER , the best performance is achieved by vGraph/vGraph+ on 5, ComGAN on 4 and ComE on 3 out of 13 datasets in terms of F1-score. A a similar trend can be seen in Jaccard Similarity. Third, it is noted that all the methods which achieve second best performance are solving the task of community detection and node representation learning jointly. This supports our claim that treating the two tasks jointly results in better performance.

Fourth, we observe that vGraph+ results are generally better than vGraph. This is because vGraph+ incorporates a regularization term in the loss function which is based on Jaccard coefficients of connected nodes as edge weights. However, it should be noted that this prepossessing step is computationally expensive for densely connected graphs.

Tab. 4 shows the results on non-overlapping community detection. First, we observe that MNMF, DeepWalk, LINE and Node2Vec provide a good baseline for the task. However, these methods are not able to achieve comparable performance on any dataset relative to the frameworks that treat the two tasks jointly. Second, VECODER consistently outperforms all the competitors in NMI and ARI metrics, except for *CiteSeer* where it achieves second best ARI. Third, we observe that GCN based models i.e. GAE, VGAE and DGI show competitive performance. That is, they achieve second best performance in all the datasets except *CiteSeer*. In particular, DGI achieves second best NMI results in 3 out of 5 datasets and 2 out of 5 datsets in terms of ARI. Nonetheless, DGI results are not very competitive in Tab. 2 and Tab. 3, showing that while DGI can be a good choice for learning node embeddings for attributed graphs with non-overlapping communities, it is not the best option for non-attributed graphs or overlapping communities.

The results for node classification are presented in Tab. 5. VECODER achieves best F1-micro and F1-macro scores on 4 out of 5 datasets. We also observe that GCN based models i.e. GAE, VGAE and DGI show competitive performance, following the trend in results of Tab. 4. Furthermore, we note that the node classification results of CommunityGan (ComGAN) are quite poor. We think a potential reason behind it is that the node embeddings are constrained to have same dimensions as the number of communities. Hence, different components of the learned node embeddings simply represent the membership strengths of nodes for different communities. The linear classifiers may find it difficult to separate such vectors.

## 4.5 HYPERPARAMETER SENSITIVITY

We study the dependence of VECODER on $\epsilon$ and $\alpha$ by evaluating on four datasets of different sizes: *fb698*($N = 61$), *fb1912*($N = 747$), *amazon1000*(N=1540) and *youtube*($N = 5346$). We sweep for $\epsilon = \{0.1, 0.2, \cdots, 0.9\}$. For demonstrating effect of $\alpha$, we fix $\epsilon = 0.3$ and sweep for $\alpha = \{0.1, 0.2, \cdots, 0.9\}$. The average results of five runs for $\epsilon$ and $\alpha$ are given in Fig. 1a and Fig. 1b respectively. Overall VECODER is quite robust to the change in the values of $\epsilon$ and $\alpha$. In case of $\epsilon$, we see a general trend of decrease in performance when the threshold $\epsilon$ is set quite high e.g. $\epsilon > 0.7$. This is because the datasets contain overlapping communities and a very high $\epsilon$ will cause the algorithm to give only the most probable community assignment instead of potentially providing multiple communities per node. However, for a large part of sweep space, the results are almost consistent. When $\epsilon$ is fixed and $\alpha$ is changed, the results are mostly consistent except when $\alpha$ is set to a low value. Eq. (11) shows that in such a case the node itself is almost neglected and VECODER tends to assign communities based upon neighborhood only, which may cause a decrease in the performance. This effect is most visible in *amazon1000* dataset because it has only 1.54 points on average per community i.e. there is a good chance for neighbours of a point of being in different communities. Therefore, only depending upon the neighbors will most likely result in poor results.

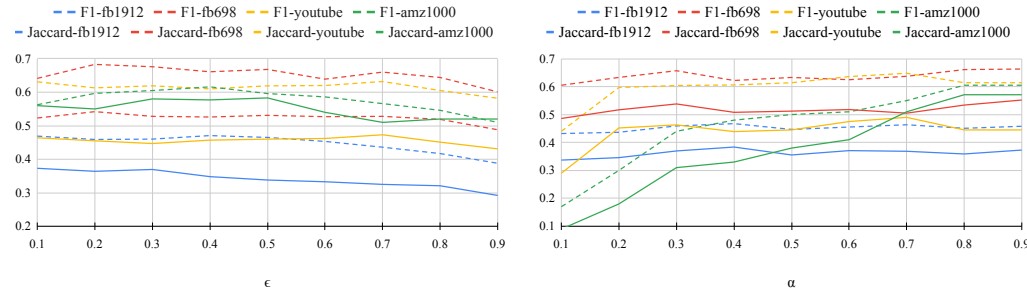

(a) Effect of $\epsilon$. Overall a slight decrease in scores can be observed after $\epsilon = 0.7$ mark.

(b) Effect of $\alpha$. The scores generally tend to decrease for small values of $\alpha$.

Figure 1: Effect of hyperparameters on the performance. F1 and Jaccard scores are in solid and dashed lines respectively.

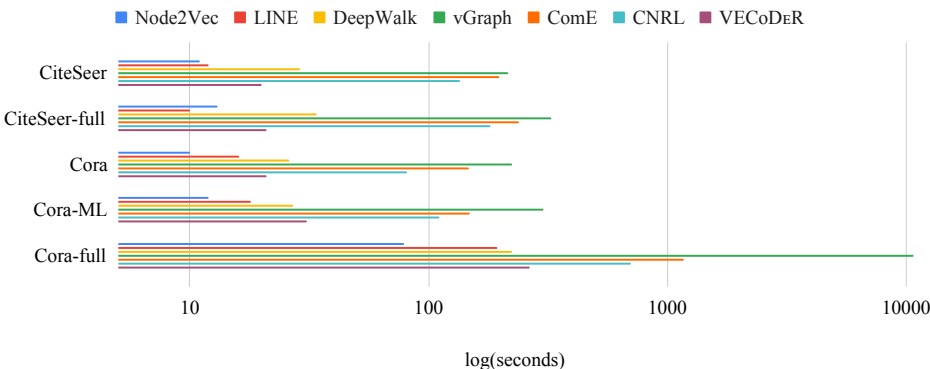

Figure 2: Comparison of running times of different algorithms.

## 4.6 TRAINING TIME

Now we compare the training times of different algorithms in Fig. 2. As some of the baselines are more resource intensive than others, we select aws instance type `g4dn.4xlarge` for fair comparison of training times. For vGraph, we train for 1000 iterations and for VECODER for 1500 iterations. For all other algorithms we use the default parameters as used in section 4.3. We observe that the methods that simply output the node embeddings take relatively less time compared to the algorithms that jointly learn node representations and community assignments e.g VECODER, vGraph and CNRL. Among these algorithms VECODER is the most time efficient. It consistently trains in less time compared to its direct competitors. For instance, it is about 12 times faster than ComE for *CiteSeer-full* and about 40 times faster compared to vGraph for *Cora-full* dataset. This provides evidence for lower computational complexity of VECODER in Section 3.5.

## 5 CONCLUSION

We propose a scalable generative method VECODER to simultaneously perform community detection and node representation learning. Our novel approach learns a single community-aware node embedding for both the representation of the node and its context. VECODER is scalable due to its low complexity, i.e. $\mathcal{O}(|\mathcal{E}|K + NKd)$. The experiments on several graph datasets show that VECODER consistently outperforms all the competitive baselines on node classification, overlapping community detection and non-overlapping community detection tasks. Moreover, training the VECODER is highly time-efficient than its competitors.

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
