# OpenReview forum: "VECoDeR - Variational Embeddings for Community Detection and Node Representation"
_ICLR.cc/2021/Conference — Reject_

### Official Review · AnonReviewer3 · 2020-10-27
**Interesting methodology, but the experimental evaluation should be enhanced.**

**Rating:** 5
**Confidence:** 5

**Review:**

The paper deals with the problem of simultaneously learning node embeddings and detecting communities on graphs. Although both tasks are particularly important while analyzing networks, most of the proposed approaches address them independently. The paper proposes a generative model, called VECODER, that aims to jointly learn overlapping communities and node representations. The proposed model follows a variational formulation which assumes that the node embeddings are generated from a prior distribution; this can be used to control how community embeddings are sampled. This leads to an encoder-decoder architecture, where the decoder ensures that similar (i.e., connected) nodes will obtain similar embeddings. The proposed model has been empirically evaluated on three tasks (overlapping and non-overlapping community detection, and node classification), and the performance has been compared against various baseline models.

Strong points:

-- The paper addresses two important problems in network analysis, namely community detection and representation learning.

-- The paper is well-structured and well-written. In particular, the part of the proposed methodology is clearly presented and overall seems to be very interesting.

-- In the experimental evaluation of the proposed model, the performance of VECODER is examined over multiple networks on three different tasks.

Weak points:

-- The main concern about the paper is related to the experimental evaluation. In particular, some important baselines methods are missing. Since the paper examines community-based embeddings, I would expect to consider M-NMF as a baseline. The authors mention M-NMF in the related work but do not consider it in the evaluation mainly due to its scalability issues. In any case, most of the datasets used in the experiments are relatively small, so I expect that M-NMF would be able to scale.

-- Besides, a few other quite important baselines are missing from both the empirical evaluation and the related work. Below, I mention three papers that are highly relevant:

- A unified framework for community detection and network representation learning, TKDD 2018
- Graph Embedding with Self-Clustering, ASONAM 2019
- CommunityGAN: Community Detection with Generative Adversarial Nets, CIKM 2019

Is there any particular reason why these models have not been considered?

-- Another point is related to the fact that the proposed model is based on an encoder-decoder framework that leverages features while learning embeddings. On the contrary, most of the used baseline models (e.g., ComE) do not take into account features while learning representations. Although node features are used while training a logistic regression classifier for node classification, to my view this is not a very fair comparison. So, my question is why the paper did not consider variants of GNN models (with unsupervised training) or Graph Autoencoders as baselines?

-- Is there any particular reason why the task of link prediction has not been considered?

-- In Table 3, in both CiteSeer networks, there is a huge gap in the performance of the proposed model compared to the baselines. Why is this happening?

-- The parameter sensitivity experiments are very interesting. Nevertheless, I would expect to also have some experiments that would demonstrate how the design choices have affected the performance of the model. For instance, as mentioned in Sec. 3.3, would there be any impact on the performance on downstream tasks if different GNN architectures are used as encoders?


Minor comments/typos:

-- In the Related work section (page 2), it is mentioned that spectral clustering is applied to the adjacency matrix for extracting communities. I would propose the authors to mention the Laplacian matrix instead.

-- Related work (page 3): some spaces are missing before references, and some others should be removed.

---

> ### Author Response · Authors · 2020-11-22
> **Response to Reviewer3**
>
> We thank the reviewer for their thoughtful comments on our work. In the light of constructive comments regarding experimental evaluation,  we ran a lot of experiments and revised the manuscript. Below we answer the questions one by one
>
> **The main concern ... M-NMF would be able to scale.**
>
> We have included M-NMF results in the revised version of the paper.
>
> ---
>
> **Besides, a few other quite important baselines are missing from both the empirical evaluation and the related work. Below, I mention three papers that are highly relevant:**
> * A unified framework for community detection and network representation learning, TKDD 2018
> * Graph Embedding with Self-Clustering, ASONAM 2019
> * CommunityGAN: Community Detection with Generative Adversarial Nets, CIKM 2019
>
> **Is there any particular reason why these models have not been considered?**
>
> We have included these algorithms in the revised version of the paper. Kindly read the paper for detailed discussion.
> * Briefly, CommunityGAN achieves second best performance on 4 out of 13 datasets on overlapping community detection task. However, for non-overlapping community detection task, it does not achieve competitive performance.
> * Furthermore, on the node classification task, the results of CommunityGAN (ComGAN) are quite poor. We think a potential reason behind it is that the node embeddings are constrained to have the same dimensions as the number of communities. Hence, different components of the learned node embeddings simply represent the membership strengths of nodes for different communities. The linear classifiers may find it difficult to separate such vectors.
> * CNRL(A unified framework for community detection and network representation learning) performs reasonably well on some datasets but only achieves competitive performance on 2 datasets on overlapping community detection and one dataset on node classification task.
> * Graph Embedding with Self-Clustering (GEMSEC) gives cluster assignments, i.e. it does not deal with the case of overlapping communities. Therefore it is included only for non-overlapping community detection and node-classification experiments. It does not achieve competitive performance on any task considered in this experimental evaluation.
>
> ---
>
> **Another point is related ... So, my question is why the paper did not consider variants of GNN models (with unsupervised training) or Graph Autoencoders as baselines?**
>
> We have included GAE/VGAE baseline results in the revised version of the paper.
>
> ---
>
> **Is there any particular reason why the task of link prediction has not been considered?**
>
> One could potentially try that. However, our intuition is that the embeddings of the members of the same community should lie closeby and the points belonging to different communities should be as separate as possible. This push-pull effect is achieved via Eq. (12). Consequently, we might have to sacrifice the few edges shared by two otherwise different communities if we are to push them apart. While this does not affect classification accuracy as demonstrated by the experiments, it can potentially affect the link prediction task. Moreover, most of the baseline methods that do clustering/community detection and node representation consider node classification to demonstrate node representation. So we adhered to node classification for sake of comparison too. Nonetheless, for future work, it could be an interesting avenue to include this task in this framework.
>
> ---
>
> **In Table 3, in both CiteSeer networks, there is a huge gap in the performance of the proposed model compared to the baselines. Why is this happening?**
>
> We have revised the results by including several strong baseline and competitive methods as pointed out by learned reviewers. We have included these 7 additional methods: NMNF, GAE, VGAE, DGI, GEMSEC, CNRL, CommunityGAN. Kindly see the revised version of the paper.
>
> ---
>
> **The parameter sensitivity experiments are very interesting. Nevertheless, I would expect to also have some experiments that would demonstrate how the design choices have affected the performance of the model. For instance, as mentioned in Sec. 3.3, would there be any impact on the performance on downstream tasks if different GNN architectures are used as encoders?**
>
> This can be an interesting avenue to explore. During our initial experiments, we tried replacing GCN encoder with an $F \times d$ matrix that simply maps input to latent space for the community detection task. The results were comparable in case of  non-overlapping case, while for the overlapping case there was a significant decrease in the performance. In the referenced code, we also provide an implementation for non-overlapping experiments using a simple $F \times d$  matrix for encoding.
>
> ---
>
> **Minor comments/typos:**
> Changed as per learned reviewer’s suggestion.

---

### Official Review · AnonReviewer2 · 2020-10-29
**Review for VECoDeR**

**Rating:** 6
**Confidence:** 4

**Review:**

(Summary)

This paper aims to learn node representations of graph to jointly satisfy node embedding properties and community detection property. Node embedding must preserve proximities guaranteeing that adjacent nodes are closer than others. Community detection must promote more similar clustering assignments to adjacent nodes than others. These two problems have been tackled separately or simultaneously but with maintaining two different node representations. The authors claim that the proposed VECoDeR is capable of learning a single community-aware node representation per node, which is jointly effective in both scenarios.


(Originality and Contributions)

Their model assumes several conditional independences such as cluster assignments of different nodes given node representation, and adjacencies on different edges given the cluster assignments and node embeddings of each end. Then the rest of them creating ELBO and using reparametrization trick for training variational autoencoder follows the standard procedure. Posterior distribution of cluster assignment in the encoder adopt the weighted message passing (similar to graph convolution or Sun et al 2019). One possibly small novelty would be the parametrization of the edge decoder. In this sense, readers must find this paper’s contribution more in the practical side – superior performance -- than in the theoretical aspects.


(Strength and weakness)
- Strength is the performance that beats all other famous models.
- Weakness is the lack of analysis that provide intuitions why and how VECoDeR outperforms other models by a big margin.


(Concerns, Questions, and Suggestions)

1) In the equation (12), the query random variable must be $a_{ij}$ rather than $e_{ij}$.

2) It would be also great to double clarify what types of and how many parameters belong to the encoder (in $\phi$) and the decoder (in $\theta$).

3) Using Gumbel-softmax trick allows you to efficiently sample one-hot vector of clustering assignment for non-overlapping community detection. For overlapping community detection, in contrast, clustering assignment matrix would be dense by assigning possibly negligible probability mass to all existing clusters. In the inference time, you are able to threshold it to reduce the dense connection by increasing $\epsilon$, but how to scale the algorithm for learning large number of clusters?

4) Related to the previous point, the number of clusters $K$ in all datasets seem to be orders of magnitudes smaller comparing to the number of nodes. This is often not satisfied in real data and some scenarios could ask overcomplete community detection (see Ananakumar et al 2012). Is the proposed model capable of detecting fine-grained community structures?

5) Similarly, the model seems to be too independent from different hyperparameter settings of $\alpha$ and $\epsilon$ than expected. At least one or two datasets where the ground-truth overlapping communities are approximately known must be tested with much larger $K$ for robustness check.

6) Assuming the reported performance are all correct, VECoDeR outperforms a variety of famous models for graph representation learning. Having some qualitative examples (than abstract explanation in high-level about structure preservation) would be beneficial. In particular, some part of the graph that does not performed well in all other models but correctly done by VECoDeR. Current draft rarely provides such intuitions to potential readers and users.

---

> ### Author Response · Authors · 2020-11-22
> **Response to Reviewer2**
>
> We thank the reviewer for their positive feedback and constructive comments. We provide our responses and hopefully they could resolve your concerns.
>
> 1. **In the equation (12), the query random variable must be $a_{ij}$ rather than $e_{ij}$**.
>
> Thanks for pointing it out. We have updated that in the revised version.
>
> ---
> 2. **It would be also great to double clarify what types of and how many parameters belong to the encoder (in $\phi$) and the decoder (in $\theta$).**
>
> $\theta$ parametrizes the generative model and $\phi$ parametrizes the variational model. We can see from Eq. 6 that the only learnable parameters in $\phi$ depend upon the choice of encoder e.g. a GCN encoder as used in VGAE or even an $F\times d$ matrix that maps input to latent space.
>
> ---
> 3. **Using Gumbel-softmax trick allows you to efficiently sample one-hot vector of clustering assignment for non-overlapping community detection. For overlapping community detection, in contrast, clustering assignment matrix would be dense by assigning possibly negligible probability mass to all existing clusters. In the inference time, you are able to threshold it to reduce the dense connection by increasing, but how to scale the algorithm for learning a large number of clusters?**
>
> The process of using Gumbel-softmax trick is the same for both overlapping and non-overlapping community detection. As for the complexity, the decoder Eq. (12) takes O(|E|) time for assignment samples drawn using Gumbel-softmax trick. So, even when the probabilities matrix is dense, computing Eq. (12) for all cases will take O(|E|) time, which still keeps us within the same order as given in section 3.5 i.e. O(|E|K+NKd).
>
> ---
>
> 4. **Related to the previous point, the number of clusters in all datasets seem to be orders of magnitudes smaller comparing to the number of nodes. This is often not satisfied in real data and some scenarios could ask overcomplete community detection (see Ananakumar et al 2012). Is the proposed model capable of detecting fine-grained community structures?**
>
> We have added two new datasets, amazon500 and amazon1000, with 500 and 1000 communities respectively. Kindly refer to the experiments section of the revised version.
>
> ---
> 5. **Similarly, the model seems to be too independent from different hyperparameter settings of and than expected. At least one or two datasets where the ground-truth overlapping communities are approximately known must be tested with much larger for robustness check.**
>
> 	* As per learned reviewer’s suggestion, we have revised the manuscript  and added the amazon1000 dataset with 1000 communities in the hyperparameter sensitivity analysis.
> 	* VECoDeR favours larger values of alpha for graph datasets having a large number of small-sized communities. Because then small values of alpha will give more weightage to the neighboring nodes that are more likely to be in different communities, when K is very large.
>
> ---
>
> 6. **Assuming the reported performances are all correct, VECoDeR outperforms a variety of famous models for graph representation learning. Having some qualitative examples (then abstract explanations in high-level about structure preservation) would be beneficial. In particular, some part of the graph that does not perform well in all other models but is correctly done by VECoDeR. Current draft rarely provides such intuitions to potential readers and users.**
>
> 	* We have revised the manuscript  and discussed in detail the insights into the problem and reasons for superior performance. Kindly read the revised manuscript.
> 	* To ensure reproducibility of the results, we publish our code here: https://anonymous.4open.science/r/1d95bf8f-8ce3-4870-a454-07db463b419f

---

### Official Review · AnonReviewer1 · 2020-10-29
**Interesting problem, but limited novelty**

**Rating:** 5
**Confidence:** 4

**Review:**

This paper proposes a generative model for community detection and node representation in a unified framework. The underlying assumption of the work is that connected nodes in a graph should have similar node embeddings and similar community assignments. Experimental evaluations show improvement both on node classification and community detection (both overlapping and non-overlapping).

The problem of integrating community detection and node representation in a unified framework is important and needs more attention from the researchers. The paper also motivates the problem well. The overall presentation is good and easy to follow. However, there are few concerns I have about the paper, as stated below.

1. The technical approach of the paper seems to be quite similar to the cited work vGraph (NeurIPS 2019). It would be good to point out the major differences between the two works.

2. Joint community detection and node representation learning - more discussion is needed for this subsection. Specifically, the existing works which integrate these two tasks and limitation of those works should be explained.

3. In Eq. 3, is it fair to assume that z_i 's are iid random variables? Embedding of a node should depend on the embeddings of neighbors at the least.

4. The GCN encoder aggregates the embeddings of the neighbors of a node while modeling q_{\phi}(z_i | I). Then what is the need of Equation 11 instead of Equation 10?

5. The last term of Equation 7 is a sum over all pairs of nodes, not only on the edges. Should not it lead to a computational complexity of O(N^2)?

6. DGI (ICLR 2019) should be used for both non-overlapping community detection and node classification, as it is a SOTA algorithm for unsupervised node representation learning.

7. In Section 4.4, only results in Tables 2-4 are presented in words. I could not see any analysis and insight in this section. The reason of superior performance of VECODER compared to individual baselines is also missing.

8. Can authors also report classification accuracy on Cora and Citeseer, as that is a standard adopted by different SOTA network representation algorithms?


------------------------------------

Authors did answer some of my doubts through their response. However, I am still not very convinced if the paper has sufficient novelty to get accepted in ICLR. I increase my score from 4 to 5.

---

> ### Author Response · Authors · 2020-11-22
> **Response to Reviewer1**
>
> We thank the reviewer for providing constructive comments.  We ran additional experiments triggered by the questions that were raised in the assessment of our manuscript and modified it accordingly.
> We reply individually to each raised point below.
>
> 1. We add the response to this in a separate comment because of 5000 characters limit for response text.
>
> ---
> 2. We have revised the manuscript and expanded the related work section of joint community detection and node representation learning by adding CNRL, ComE,  GEMSEC, CommunityGAN, and vGraph.  We also include all of these methods in our experimental evaluation and discuss their limitations.
> ---
> 3.  VECoDeR relies on the decoder to ensure the dependence of connected nodes and their communities with each other. The approach to assume that z_i 's are iid random variables is commonly followed, e.g. in VGAE [https://arxiv.org/pdf/1611.07308.pdf],  NetVAE[https://www.ijcai.org/Proceedings/2019/0370.pdf], EVGAE [https://arxiv.org/pdf/2004.01468.pdf].
> ---
> 4. The framework proposed in this work proposes VECoDeR with several design choices for encoder. Eq. 11 explicitly ensures that the embedding of a community depends not only upon a node but also its neighbors. Even when GCN is used for encoding, Eq. 11 allows us to control the bias between the effect of node versus the effect of its neighbors on the community embedding.
> ---
> 5. The summation is over the edges just as done in GAE/VGAE. This avoids the complexity of O(N^2). We have explicitly mentioned that in the revised version.
> ---
> 6. DGI has been included in the revised version of the manuscript. Kindly read the experiment section and discussion of results.
> 	* It performs well in datasets with non-overlapping communities. However, It is not very competitive in overlapping community detection. This shows that while DGI can be a good choice for learning node embeddings for attributed graphs with non-overlapping communities, it is not the best option for non-attributed graphs with overlapping communities. VECoDeR, on the other hand, performs consistently well in both scenarios.
> 	*Even for non-overlapping community detection task, DGI performs quite poor on Cora-full dataset. This is an interesting result and it warrants further investigation. Our intuition is that since DGI discriminates by permuting node embeddings, it might be less competitive for the graphs with high edge density, compared to the algorithms that explicitly model edge information. That might be a reason why it performs poor on Cora-full, which has higher edge density compared to other non-overlapping datasets.
> ---
> 7. As per learned reviewer’s suggestion, we have revised the manuscript  and discussed in detail the insights into the problem and gave analysis on the performance of methods. Kindly read the revised manuscript.
> ---
> 8. Classification accuracy is already reported on Cora and Citeseer.
> In multiclass classification tasks for which every test case is guaranteed to be assigned to exactly one class, micro-F1 is equivalent to accuracy. It won't be the case in multi-label classification.

---

> > ### Author Response · Authors · 2020-11-23
> > **Response to Reviewer1 Concern (1) - major differences from vGraph**
> >
> > We have revised the manuscript to highlight the differences with vGraph.
> > We also provide a more detailed account here:
> > * vGraph learns two embeddings for each node: one used for the node representation task and the second “context” embedding aids in community detection. vGraph requires two node embeddings for parameterizing two different distributions. In contrast, we propose learning a single community-aware node representation which is directly used for both tasks. In this way, we not only get rid of an extraneous node embedding but also reduce the computational cost.
> > * The variational models of vGraph and VECoDeR are quite different. vGraph assumes that each node can be represented as a mixture of multiple communities and is described by a multinomial distribution over communities, whereas VECoDeR models the node embedding by a single distribution (std. Gaussian in our case).
> > * For a given node, vGraph, first draws a community assignment and then a connected neighbor node is generated based on the assignment. Whereas, VECoDeR draws the node embedding from prior distribution and then community assignment is conditioned on a single node only. In simple terms, vGraph also needs edge information in the generative process whereas VECoDeR does not require it. VECoDeR relies on the decoder to ensure that embeddings of the connected nodes and their communities share high similarity with each other.
> > * From a data representation perspective, the two node embeddings learnt by vGraph give two disparate perspectives into the data, e.g. two nodes might be closer in one embedding space compared to the other. VeCoDeR gives a uniform view into the data through a single embedding.
> > * For large datasets, the training of the generative model of vGraph needs a large number of samples to approximate the distribution of nodes given a community [see: vGraph  Eq. (3)]. On the other hand, VECoDeR generative process is simpler and straight-forward to implement and learn.
> > * vGraph+ results are generally better than vGraph for overlapping community detection. This is because vGraph+ incorporates a regularization term in the loss function which is based on Jaccard coefficients of connected nodes as edge weights. However, it should be noted that this prepossessing step is computationally expensive for densely connected graphs. VECoDeR, on the other hand, has less complexity and does not force any such preprocessing step.

---

### Official Review · AnonReviewer5 · 2020-11-12

**Rating:** 5
**Confidence:** 5

**Review:**


In this work the authors propose a generative model for jointly learning a representation of nodes and communities in a joint framework.  The authors perform a wide variety of experiments on community (overlapping, non-overlapping) detection, and noed classification. This paper is potentially very interesting due to the large improvements it shows on some important tasks, but there are several non-trivial issues that I discuss in the following, and due to which I feel the paper is not above the bar in its current form.

1. The authors should clarify early on, whether the focus is on undirected unweighted graphs, or general directed weighted graphs.

2. The authors mention "In the literature these tasks are treated separately". The authors should discuss the problem more precisely. For instance if I have two connected components, each being a clique, most node embedding methods will capture this by placing nodes from each clique on the same point (or really close). If the two connected components are sparse, the nodes from each component will be embedded closer to each other compared to nodes from the other connected component.  So, standard graph embedding methods do capture basic connectivity, i.e., crude connectivity structure. This bring me also to the next point.
3. How is the notion of a community defined here? Do you consider a community a set of low conductance? Or some set of nodes that share certain common attributes? I assume the former, i.e., some form of a well-connected set of nodes that is less well-connected to the rest of the graph.
4. Can the authors comment on the improvements they observe using their method? Is it due to the fact that labels within the community are overwhelmingly similar?  The authors use as competitors methods that do not take into account features. If the difference lies in this, the authors should compare also against graph convolutional networks. In any case, a more detailed explanation of the improvements is required in my opinion. See also point 8 for an important reference to this work.
5. In terms of competitors for overlapping community detection the authors should take a look at  the recent paper "SimClusters: Community-Based Representations for Heterogeneous Recommendations at Twitter" that describes some mechanics of Twitter's recommendation system, and that uses the objective, and a variation of the algorithm  proposed in "Provably fast inference of latent features from networks: With applications to learning social circles and multilabel classification" WWW 2015 for overlapping community detection. The algorithms in these two papers for instance outperform BigClam.
6. Does the proposed method yield any concrete theoretical improvement? For example, have the authors tried to show that it achieves better than Cheeger's approximation guarantee? Perhaps some simulation results on synthetic dataset can provide at least some experimental evidence by looking inner products of nodes within and across communities.
7. It would be helpful to explain the novelty of this work compared to the vGraph Neurips 2019 paper.
8. Concerning the top 5 communities choice, the authors should also take a look at this paper  "The ground truth about metadata and community detection in networks" Science Advances  where the notion of ground-truth used in this work is questioned.

---

> ### Author Response · Authors · 2020-11-22
> **Response to Reviewer5-part1**
>
> We thank the reviewer for their thought-provoking feedback and constructive criticism.  We provide our responses and hopefully they can resolve your concerns.
>
> 1. We focus on undirected graphs. We have mentioned it explicitly in the revised paper as per the suggestion of the learned reviewer.
> 2. The authors agree with the learned reviewer comment that standard graph embedding methods capture basic connectivity. That is why we included baseline methods like DeepWalk, node2vec, LINE etc. From these learned embeddings, we can get community assignments e.g. by using KMeans/GMM etc. However, these community assignments are usually subpar compared to the algorithms that learn node embeddings and community assignments jointly (e.g. ComE, VECoDeR). Looking from the other perspective, methods like CommunityGAN/BigClam that focus primarily on finding the community structure in the dataset, perform poorly for node-representation tasks e.g. node classification. That’s why, it is worth exploring the approaches that jointly learn community-aware node embedding.  We have included it explicitly in the revised paper as per the suggestion of the learned reviewer.
> 3. Yes, we consider community as “some form of a well-connected set of nodes that is less well-connected to the rest of the graph”. Our intuition is to maximize the number of intra-community edges and minimize the number of inter-community edges. This is achieved by pulling the connected nodes to the same community and pushing apart the communities of unconnected nodes as done in Eq. (12).
> 4.
> 	* The availability of features is briefed in Table 1.
> 	* For the datasets, where VECoDeR uses features, all the competitors methods also use them.
> 	* For the algorithms that by design do not use node features, we train the classifier by appending the raw node features with the learnt embeddings. Moreover, we have revised the results by including several strong baseline and competitive methods as pointed out by learned reviewers, including GAE, VGAE and DGI which use GCN encoders for node embeddings.
> 	* It is to be noted that VECoDeR learns general node embeddings which can be then used for a variety of tasks. This is in contrast to some methods like GCN [Kipf & Welling 2016] which learn classification-specific node embeddings by training the graph itself with a classification loss.In other words, labels are not required by the VECoDeR for learning the node embedding, and used only for training the logistic regression classifier, when doing node classification.
> 5.
> 	* Before answering this comment, we would like the reviewer to kindly see the revised version of the manuscript. We have compared the performance with 11 methods for overlapping community detection and 13 methods for non-overlapping case. Four of these competitive methods were published in 2019.
> 	* The paper SimClusters got published in August 2020 and our submission was in September 2020 i.e. the two works were done in parallel and therefore it was really not possible for us to consider it in our experiments. After your comment, we looked for it. They have only published part of their code i.e. 1 stage out of 3 stages of the algorithm.
> 	* The paper “Provably fast inference ...” might be a very good paper, but it was not included by any of the baselines and competitive methods (ComE, vGraph etc) that we mentioned in our work. Moreover, the suggested paper is barely cited in the literature, making it rather hard for us to dig it. Also, unfortunately, the authors do not offer the implementation of the algorithm, which makes including the paper in experiments quite difficult.
> 	* Just a comment on BigClam: this algorithm is not among the top 3 methods in our experiments. This shows the competitive nature of the included baselines.
>
> 6. This can be an interesting future work to dive deeper into the theoretical analysis. We acknowledge that the Cheeger inequality has been very useful in many applications of random walk type problems for bounding the eigenvalues of the graph. Unfortunately, such an analysis is not in the scope of our current work. In this work, we have proposed a variational model VECoDeR for joint community detection and node representation learning. This model not only outperforms the current SOTA methods for the joint task but also is time-efficient in comparison to its direct competitors.
>
> 7. We add the response to this in a separate comment because of 5000 characters limit.
>
> 8. That is an excellent paper and raises some important questions regarding metadata and community detection in networks.
> However, in order to compare our results with competitors methods we employed the evaluation metrics most commonly used by the competitors and baselines.

---

> > ### Author Response · Authors · 2020-11-22
> > **Response to Reviewer5-novelty compared to the vGraph**
> >
> > 7. We have revised the manuscript to highlight the differences with vGraph.
> > We also provide a more detailed account here:
> > 	* vGraph learns two embeddings for each node: one used for the node representation task and the second “context” embedding aids in community detection. vGraph requires two node embeddings for parameterizing two different distributions. In contrast, we propose learning a single community-aware node representation which is directly used for both tasks. In this way, we not only get rid of an extraneous node embedding but also reduce the computational cost.
> > 	* The variational models of vGraph and VECoDeR are quite different. vGraph assumes that each node can be represented as a mixture of multiple communities and is described by a multinomial distribution over communities, whereas VECoDeR models the node embedding by a single distribution (std. Gaussian in our case).
> > 	* For a given node, vGraph, first draws a community assignment and then a connected neighbor node is generated based on the assignment. Whereas, VECoDeR draws the node embedding from prior distribution and then community assignment is conditioned on a single node only. In simple terms, vGraph also needs edge information in the generative process whereas VECoDeR does not require it. VECoDeR relies on the decoder to ensure that embeddings of the connected nodes and their communities share high similarity with each other.
> > 	* From a data representation perspective, the two node embeddings learnt by vGraph give two disparate perspectives into the data, e.g. two nodes might be closer in one embedding space compared to the other. VeCoDeR gives a uniform view into the data through a single embedding.
> > 	* For large datasets, the training of the generative model of vGraph needs a large number of samples to approximate the distribution of nodes given a community [see: vGraph  Eq. (3)]. On the other hand, VECoDeR generative process is simpler and straight-forward to implement and learn.
> > 	* vGraph+ results are generally better than vGraph for overlapping community detection. This is because vGraph+ incorporates a regularization term in the loss function which is based on Jaccard coefficients of connected nodes as edge weights. However, it should be noted that this prepossessing step is computationally expensive for densely connected graphs. VECoDeR, on the other hand, has less complexity and does not force any such preprocessing step.

---

### Decision · Program_Chairs · 2021-01-07
**Final Decision**

**Decision:**

Reject

**Comment:**

The paper proposes a new method to learn representation and community structure of a network jointly. The reviewers agree that the paper contains some interesting ideas but they raise also some important concerns. For example:

- even after considering the authors' rebuttal, the paper seems not too novel. In particular the results seem a bit incremental over vGraph.

- the notion of community is not formalized by the authors neither in the paper or in the rebuttal. The paper would benefit greatly by having such formal definition

Overall, the paper is interesting but it does not meet the high acceptance bar of ICLR